# Transcription Factors Associated with Abiotic and Biotic Stress Tolerance and Their Potential for Crops Improvement

**DOI:** 10.3390/genes10100771

**Published:** 2019-09-30

**Authors:** Elamin Hafiz Baillo, Roy Njoroge Kimotho, Zhengbin Zhang, Ping Xu

**Affiliations:** 1Key Laboratory of Agricultural Water Resources, Hebei Laboratory of Agricultural Water Saving, Center for Agricultural Resources Research, Institute of Genetics and Developmental Biology, University of Chinese Academy of Sciences, Shijiazhuang, Hebei 050021, China; aminomooon14@gmail.com (E.H.B.); kimroybiotech@gmail.com (R.N.K.); xuping@sjziam.ac.cn (P.X.); 2University of Chinese Academy of Sciences, Beijing 100049, China; 3Innovation Academy for Seed Design, Chinese Academy of Sciences, Beijing 100101, China; 4Agricultural Research Corporation (ARC), Ministry of Agriculture, Gezira 21111, Sudan

**Keywords:** transcription factors, abiotic and biotic stress response, overexpression, gene, sorghum

## Abstract

In field conditions, crops are adversely affected by a wide range of abiotic stresses including drought, cold, salt, and heat, as well as biotic stresses including pests and pathogens. These stresses can have a marked effect on crop yield. The present and future effects of climate change necessitate the improvement of crop stress tolerance. Plants have evolved sophisticated stress response strategies, and genes that encode transcription factors (TFs) that are master regulators of stress-responsive genes are excellent candidates for crop improvement. Related examples in recent studies include TF gene modulation and overexpression approaches in crop species to enhance stress tolerance. However, much remains to be discovered about the diverse plant TFs. Of the >80 TF families, only a few, such as NAC, MYB, WRKY, bZIP, and ERF/DREB, with vital roles in abiotic and biotic stress responses have been intensively studied. Moreover, although significant progress has been made in deciphering the roles of TFs in important cereal crops, fewer TF genes have been elucidated in sorghum. As a model drought-tolerant crop, sorghum research warrants further focus. This review summarizes recent progress on major TF families associated with abiotic and biotic stress tolerance and their potential for crop improvement, particularly in sorghum. Other TF families and non-coding RNAs that regulate gene expression are discussed briefly. Despite the emphasis on sorghum, numerous examples from wheat, rice, maize, and barley are included. Collectively, the aim of this review is to illustrate the potential application of TF genes for stress tolerance improvement and the engineering of resistant crops, with an emphasis on sorghum.

## 1. Introduction

Abiotic and biotic stresses are major environmental threats that greatly reduce crop yield. Examples of abiotic stresses include drought, salt, cold, and heat, and biotic stresses include diverse living organisms, such as fungi, bacteria, viruses, nematodes, and insects [1,2]. Plants live continually with these stresses, giving rise to complex response interactions, but nevertheless crop productivity is severely impacted. For instance, abiotic stresses can reportedly cause yield losses of more than 50% [3], and the yield loss due to biotic stresses has been estimated as ~35 % [4]. Considering future climate change scenarios and harsh weather conditions, there is an urgent need to better understand the complex responses of plants to single and combined stresses, to ultimately enhance crop tolerance to changing climate conditions.

Cereals include more than 10,000 species worldwide, and the following five cereal crops provide much of the world’s food [5]: wheat (*Triticum sp.*), rice (*Oryza sativa*), maize (*Zea mays*), barley (*Hordeum vulgare*), and sorghum (*Sorghum bicolor*). Among cereals, sorghum is the fifth most important crop and has unique adaptations that allow it to withstand harsh conditions at different growth stages [6,7]. Sorghum is a C4 species with a small genome size (730 Mb), making it an ideal model for genomic and functional studies of plant growth under stress conditions [8]. Because it thrives under water scarcity and high temperatures, it is also an excellent model species for transcription factor studies aimed at enhancing tolerance or resistance to stresses, particularly drought [9]. Plant response strategies to abiotic and biotic stresses involve changes at the molecular, cellular, biochemical, and physiological levels. These various responses are generally controlled by several key genes encoding transcription activators and repressors that regulate downstream stress-induced genes and pathways. In the last decade, extensive research has helped identify key factors involved in abiotic and biotic stress responses [10].

Transcription factors (TFs) are central regulators of gene expression, and as such they modulate essential aspects of plant function, including responses to environmental factors and hormones, and cell differentiation and organ development [11]. TFs modulate gene expression by binding to local and distal *cis*-elements of a given gene under different biological contexts. Inukai et al. [12] highlighted recent findings that elaborate how TF interactions, local DNA structure, and genomic features can influence TF binding to DNA. Central to the function of TFs is their ability to bind specific DNA sequences and interact with different proteins in transcriptional complexes that regulate the expression of a vast number of genes. Accordingly, deciphering the mechanistic actions of TFs is essential for future studies. In plants, ~10% of genes encode TFs [13], which take part at different stages for a specific function. Several TF databases are now available and provide comprehensive information about TF families in different species (Table 1). In plant transcription factor database (plant TFDB v5.0, Center for Bioinformatics, Peking University), 134 WRKY, 180 NAC, 145 MYB, 172 ERF, and 166 bZIP genes have been identified in sorghum (Figure 1). TFs are promising candidates for genetic engineering due to their roles as master regulators of several stress-related genes. Many TF families, including WRKY, MYB, NAC, and bZIP (Table 2), have been implicated in stress responses, and many TF genes are associated with enhanced tolerance in both model and crop plants [14]. In Arabidopsis, ~30 TF families are predicted to include 1922 TFs involved in different functions [8]. A total of 2448 TFs has been reported in sorghum, along with 1611 in rice and 3337 in maize [15]. In the last two decades, a wealth of studies have identified TF genes and characterized their responses to abiotic and biotic stresses. The improvement of plant stress tolerance by manipulating the expression of TF genes has become a hot area of research, as many of these genes are stress-responsive and govern a plethora of downstream genes. Accordingly, there is potential to engineer crops with higher stress tolerance [16]. In this context, considerable progress has been made through the overexpression of several TF genes.

This review summarizes recent studies on the major TF families involved in abiotic and biotic stress responses and their potential to boost stress tolerance in sorghum. Recent studies of TFs in major cereal crops (wheat, rice, maize, and barley) are discussed; also, TFs involved in stress responses in sorghum are highlighted. Other recent reviews about stress response TFs in other crops include one by Kimotho et al. [17], which covers maize TFs involved in abiotic stress, and another by Hong et al. [18], which provides a comprehensive update on wheat TFs involved in defense responses against pathogen infection.

## 2. The Regulatory Functions of Plant TFs in Response to Abiotic and Biotic Stresses

Plants have evolved rapid response strategies to unfavorable conditions, and these involve interconnected networks at the molecular level controlled by signal cascades. The different components of stress responses are signal perception, signal transduction, and expression of stress-responsive genes [19]. When plant cells perceive a stress signal, receptors or sensors in the cell wall or membrane detect the stress stimulus, followed by a rapid response that transduces the external signal to intracellular signals. Signal cascades involving intracellular molecules or ions are activated along with kinase cascades, which are generally cytoplasmic. Major cascades are associated with reactive oxygen species (ROS) and calcium ions (Ca^2+^). Phytohormones, including abscisic acid (ABA), jasmonic acid (JA), salicylic acid (SA), and ethylene (ET), are powerful second messengers that coordinate signal transduction pathways during stress responses. These signals activate several parallel transduction pathways, which often involve phosphatases and protein kinases [19]. Following the initial step of signal perception, plants activate two major signal cascades: the mitogen-activated protein kinase (MAPK) and calcium-dependent protein kinase (CDPK) pathways [10,20]. Finally, specific TFs are upregulated or downregulated by protein kinases or phosphatases, and the TFs bind to *cis*-elements of stress-related genes to enhance or suppress their transcription. Figure 2 broadly summarizes the general activation and activity of TFs. Because many interactions establish the regulatory networks that modulate the expression of stress-responsive genes, the functions of TFs in plants are tremendous in scope. Fundamentally, however, TFs modulate the expression of key downstream genes. The following sections summarize recent findings on the major TF families.

## 3. NAC TFs

The NAC (NAM, CUC, and ATAF) TF family is one of the largest plant-specific TF families [21]. The first reported NAC proteins include NAM from petunia (*Petunia hybrida*) [22] and *Arabidopsis thaliana* Ataf1/2 and Cuc2 proteins [23]. NAC TFs are characterized by a highly conserved DNA-binding NAC domain (~150 amino acids) in the N-terminal region. In contrast, their C-terminal transcription regulatory (TR) domains are diverse. These NAC protein TR domains contain a transmembrane domain that can either activate or repress transcription [24]. Previous studies have shown that NAC proteins act via an ABA-dependent pathway as well as an ABA-independent pathway, and play a vital role in both abiotic and biotic stresses [25]. Genome-wide studies and expression analyses have identified *NAC* genes in different species; for example, there are 117 nonredundant *NAC* genes in Arabidopsis and 151 in rice [26], and 87 *NAC* genes were recently reported in sesame [27]. Functional studies have shown that *NAC* genes can be induced by different abiotic stresses, such as the induction of Arabidopsis *ANAC055* expression by drought, *ANAC072* by ABA, and *ANAC019* by salinity. Overexpression of these three genes in Arabidopsis resulted in improved drought tolerance [28,29]. *NAC* genes have also been found to regulate defense responses to pathogen invasion, wounding, and insect feeding [24].

Genome-wide analysis in sorghum identified 145 *NAC* genes, including those potentially involved in abiotic stress responses. Seven stress-related *NAC* genes (*SbNAC6*, *SbNAC17*, *SbNAC26*, *SbNAC46*, *SbNAC56*, *SbNAC58*, and *SbNAC73*) had varying expression levels over time in response to various stresses (salinity, cold, ABA, and dehydration), indicating the potential functions of *SbNAC* genes under abiotic stress conditions [30]. A separate systematic analysis of the sorghum genome [31] identified 131 *SbNAC* genes encoding 183 proteins. The authors characterized 13 *SbNAC* genes as responsive to abiotic stress. qRT-PCR expression profiling of two cultivars, one tolerant and one susceptible, showed that eight *SbNAC* genes acted as positive (*SbNAC014*, *SbNAC034*, *SbNAC035*, *SbNAC037*, and *SbNAC041*) or negative (*SbNAC052*, *SbNAC073*, and *SbNAC116*) regulators in response to post-flowering drought stress in sorghum. Moreover, under this stress condition, the expression levels of some *SbNAC* genes were significantly related to grain yield, indicating their involvement in yield maintenance and drought tolerance [30]. The abiotic stress response gene *SbSNAC1* is reportedly induced by drought, salinity, and ABA, with *SbSNAC1* overexpression leading to enhanced drought stress tolerance in transgenic Arabidopsis plants. Interestingly, the transactivation activity of *SbNAC1* is associated with its C-terminal region rather than N-terminus NAC domain [32]. *SbSNAC1* is closely related to maize *ZmSNAC1* and rice *SNAC1*, and all three genes have been shown to play a crucial role in abiotic stress responses [33].

Recent studies have shown that NAC TFs are also involved in biotic stress responses. For example, Zhang and Huang. [34] reported the induction of NAC TFs in response to greenbug infestation in sorghum. RNA-sequencing analysis indicated that many *NAC* genes were expressed in response to greenbug attack. The authors concluded that *SbNAC* genes have diverse functions in response to greenbug infestation and may therefore contribute to genetic resistance to greenbug.

In wheat, Zhang et al. [35] reported that the *TaNAC47* gene is induced by cold, salt, drought, and ABA, based on differential expression levels. *TaNAC47* overexpression in transgenic Arabidopsis led to ABA hypersensitivity and enhanced tolerance to salt, drought, and freezing stresses. Interestingly, *TaNAC47* overexpression also led to several physiological and biochemical changes, such as changes in the content of soluble sugars and proline, due to the activation of downstream genes such as *AtRD29A*, *AtRD29B*, and *AtP5CS1*; these changes may have enabled the transgenic plants to overcome the stress conditions. Huang et al. [36], also using a transgenic plant approach, found that overexpression of wheat *TaNAC29* enhanced drought and salt tolerance in transgenic Arabidopsis plants, indicating the involvement of *TaNAC29* in the response to these particular stresses. Moreover, ABA signaling and antioxidant enzymes were found to participate in the *TaNAC29*-induced stress tolerance processes. Saad et al. [37] improved wheat tolerance to drought and salt stress by introducing the rice stress response *NAC* gene *SNAC1* into wheat under the control of a maize ubiquitin promoter, based on the previous finding that *SNAC1* could induce abiotic stress tolerance in rice. Specifically, the transgenic wheat expressing rice *SNAC1* had significantly enhanced salinity and drought tolerance over many generations. qRT-PCR results revealed that *SNAC1* regulated other genes involved in abiotic/ABA signaling and regulatory components of the ABA receptor. In addition to these responses, wheat *NACs* have also been implicated in the response to powdery mildew disease. For example, *TaNAC6* overexpression reduced the haustorium quantity of the pathogen and enhanced resistance to the pathogen in transgenic plants [38].

In rice, there are a considerable number of *NAC* genes involved in response to abiotic as well as biotic stresses. Zhou et al. [38] reported that rice stress-responsive *SNAC3* was induced by drought, high temperature, ABA, and salinity stress. In transgenic plants, *SNAC3* overexpression improved tolerance to drought, high temperature, and oxidative treatment using methyl viologen. In contrast, suppression of the *SNAC3* gene by RNA interference (RNAi) increased sensitivity to these abiotic stresses. Strikingly, transgenic plants overexpressing this gene had lower levels of malondialdehyde, hydrogen peroxide (H_2_O_2_), and relative electrolyte leakage compared to the wild type control under heat stress treatment. The authors proposed that *SNAC3* may confer tolerance to these stresses by modulating ROS homeostasis. Overall, the study demonstrated the crucial role of *SNAC3* in response to different stresses and highlighted its potential for engineering crops with improved tolerance to stress such as drought and heat [39]. In addition, overexpression of the *ONAC045* gene enhanced drought and salt stress tolerance in transgenic rice plants, consistent with the previous finding that the gene was induced by drought, high salt, low temperature, and ABA stress [40].

In a recent study, four rice NAC genes, *ONAC5/6/9*, and *ONAC10* were overexpressed in rice, which resulted in improved drought tolerance and reduced grain loss under drought stress conditions compared to wild type plants [41]. The authors concluded that ONACs act as the cellular component that regulate many target genes such as TF genes, transmembrane/transporter genes, and auxin/hormone-related genes, which could potentially alter root architecture for drought tolerance [41]. Similarly, *ONAC14* overexpression in rice improved drought tolerance, and the transgenic plants had a higher panicle number and filling rate under drought conditions. The overexpression analysis indicated that *ONAC14* may mediate drought tolerance through DNA repair and defense responses, based on the findings in transgenic rice [42]. In barley, Chen et al. [43] used the transgenic approach to investigate the functions of *HvNAC6* in response to *Blumeria graminis* f. sp. *hordei* (Bgh) with respect to the ABA phytohormone. *HvNAC6* silencing by RNAi reduced *HvNAC6* transcript levels, and the plants were more susceptible to the Bgh pathogen compared to wild type plants.

In contrast to many of the above examples, NAC TFs can also downregulates some stress-related genes. For instance, *OsNAC2* overexpression in rice reduced resistance to drought and high salt stress [44]. The study showed that ABA-dependent stress-related genes were downregulated in *OsNAC2* overexpression lines. *OsNAC2* binds to the *cis*-element of the stress-related marker genes *STRESS-ACTIVATED PROTEIN KINASE1(OsSAPK1)* and *LATE EMBRYOGENESIS ABUNDANT-3* (*OsLEA*3) in the abiotic stress and ABA pathways [44]. Focusing on the mechanisms of NAC-mediated disease responses in rice, Liu et al. [45] reported that *ONAC66* was significantly induced by blast disease infection. *ONAC66* overexpression in rice enhanced resistance to bacterial blight and blast disease, and this enhancement was accompanied by suppressed expression of ABA-related genes. These findings clearly suggested that *ONAC66* modulates ABA signaling pathway components to promote disease resistance.

Reports on stress-responsive *NAC* genes show that they can respond to different stresses in different plant species “systems”. Transgenic rice overexpressing the *NAC67* gene from finger millet (*Eleusine coracana*) had enhanced salinity and drought stress tolerance under greenhouse conditions [46]. Specifically, *EcNAC67* overexpression in rice helped maintain a high water content and sustainable grain yield under drought conditions, highlighting the potential of using *EcNAC67* as a novel gene to improve drought and salinity stress tolerance in rice and other crops. Transgenic Arabidopsis overexpressing maize *NAC55* had enhanced drought resistance compared to wild type [47]. Similarly, overexpression of *NAC57* from poplar in Arabidopsis plants significantly improved tolerance against salt stress [48].

In conclusion, NAC TFs have been shown to play crucial roles in response to both abiotic and biotic stress. In sorghum, some efforts have been made to decipher the role of *NAC* genes through genome-wide analysis and expression analysis. However, compared to other cereal crops such as wheat, rice, and maize, there has been less progress in sorghum. For example, in wheat and rice, NAC TFs have been reported extensively. Transgenic approaches have underscored the potential of *NAC* genes for genetic engineering aimed at enhancing stress tolerance in sorghum and other economically important crops.

## 4. MYB TFs

Two decades ago, the *COLORED1* (*C1*) gene from *Zea mays* was the first identified plant *MYB* gene, and it encodes a MYB domain protein that is essential for anthocyanin biosynthesis in the aleurone of maize [49]. MYB TF characterization is based on the presence of highly conserved MYB domains involved in DNA binding. Generally, these domains contain multiple repeats (R); each repeat corresponds to approximately 52 amino acids and three alpha helices, the second and third of which form a helix–turn–helix (HTH) structure [50]. MYB TFs are classified into four groups according to the number of adjacent repeats in their MYB domains: 1R-MYB (one repeat), R2R3-MYB (two repeats), 3R-MYB (three repeats), and 4R-MYB (four repeats). R2R3-MYB proteins are abundant and specific to plants, with ~100 R2R3-MYBs in monocot and dicot genomes. R2R3-MYBs play vital roles in plants, including abiotic and biotic stress response [50]. In foxtail millet, rice, and Arabidopsis, the numbers of MYB members are 209, 183, and 198, respectively [51,52].

MYB functions have been extensively investigated in different plant species, and an extensive review by Ambawat et al. [53] described the roles of MYB TFs in different plant processes, including abiotic and biotic stress response. Another review by Dubos et al. [50] focused on the roles of MYB TFs in Arabidopsis, thoroughly covering their functional characterization and describing how MYBs are key factors in regulating abiotic and biotic stress responses. Given the abundance of research in this area, it is worth highlighting how the functions associated with MYB TFs will help improve stress tolerance in economically important crops. For example, as reviewed by Fang et al. [39], MYB TFs are active factors in abiotic stress signaling; MYBs have been found to regulate downstream genes in response to abiotic stresses, and they potentially act at both the transcriptional and post-transcriptional level. Last but not least, the authors of [54] reviewed the roles of MYB TFs in drought response mechanisms, providing specific examples of MYB functions and discussing the potential application of MYBs.

In sorghum, MYB TFs have been shown to play essential roles in response to both abiotic and biotic stresses. For instance, sorghum responds to attack by the fungal pathogen *Colletotrichum sublineolum* by generating 3-deoxyanthocyanidin phytoalexins, whose biosynthesis requires the sorghum yellow seed 1 (Y1) MYB TF [54]. Interestingly, transgenic maize expressing this MYB TF gene had induced 3-deoxyanthocyanidin, which enhanced resistance against leaf blight pathogen ingress [55]. Although MYB TFs have been linked to abiotic stress responses in several monocot species, none or few are linked to these stresses in sorghum. Other studies have focused on MYB-regulated biosynthetic pathways, such as that of monolignol. Scully et al. [56] reported that *SbMYB60* overexpression activated monolignol biosynthesis in drought-tolerant sorghum. MYB TF genes have also been an area of focus in other cereals. In a study about MYB TF functions in drought stress tolerance in rice [57], the authors reported that a novel gene, *OsMYB48-1*, was induced by ABA, H_2_O_2_, dehydration, and PEG, and it was slightly expressed under salt and cold stress treatment. Furthermore, transgenic rice plants overexpressing *OsMYB48-1* exhibited tolerance to drought and salinity stress. Other findings indicated that the stress response roles of *OsMYB48-1* reflect its regulation of early ABA signaling genes, such as *OsPP2C68* and *OSRK1*; late-response genes, such as *RAB21*, *OsLEA3*, *RAB16C*, and *RAB16D*; and ABA-biosynthesis genes, such as *OsNCED4* and *OsNCED5*. In another study, transgenic maize overexpressing *OsMYB55* had improved tolerance to drought and high temperature, which was accompanied by the upregulation of several stress-related genes, based on RNA-sequencing results [58]. More recently, Tang et al. [59] found that *OsMYB6* overexpression in rice enhanced tolerance to salt and drought stress compared to wild type plants.

In maize, 22 *MYB* genes were linked to one or more abiotic stresses, with 16 *MYB* genes shown to be induced by at least two different stress treatments [60]. Accordingly, these genes may be major factors in the cross-talk between different signal transduction pathways induced by abiotic stresses. *ZmMYB30* was induced by four stress treatments, and transgenic Arabidopsis expressing *ZmMYB30* had enhanced salt tolerance and increased expression levels of several stress-related genes [59]. Overexpression of *ZmMYBIF35* in Arabidopsis improved tolerance to chilling stress, and the gene was found to regulate a number of stress-related genes under chilling and oxidative stress conditions [61].

Genome-wide analysis in foxtail millet identified 209 *SiMYB* genes, and expression profiling for 11 *SiMYB* genes revealed significant changes in their expression patterns under different abiotic and hormone stress treatments [51]. This study provides a great foundation for future research concerning the functions of *SiMYB* genes in response to environmental stresses. In other species, MYB TFs have also been implicated in the response to pathogen attack. In wheat, Shan et al. [62] found that overexpression of the wheat R2R3-MYB TF gene *TaRIM1* increased resistance against *Rhizoctonia cerealis* infection. The authors also showed that *TaRIM1* modulated defense gene expression in transgenic wheat plants by binding to the MYB-binding site. An example of MYB TF involvement in the response to insect infestation has been reported in chrysanthemum. The *CmMYB19* gene was found to be induced by aphid infestation, and overexpressing *CmMYB19* limited aphid proliferation via lignin accumulation in the transgenic plants [63].

In summary, MYB TFs play crucial roles in response to numerous environmental stresses, and the findings from different species have shed light on the how that underlie *MYB* genes are involved in abiotic and biotic stress responses. Although several *MYB* genes have been functionally characterized in model and non-model species, the characterization of these TF genes in sorghum is limited. Genome-wide identification of MYB TFs in sorghum will be critical for understanding their stress-related functions in sorghum and for developing improved varieties.

## 5. WRKY TFs

WRKY TFs compose one of the largest families of transcriptional regulators, and they regulate diverse processes in plants. Since the first WRKY TF was identified from sweet potato (*Ipomoea batatas*; *SPF1*) 25 years ago [64], extensive studies have uncovered many features of this TF family. WRKY TFs consist of ~60 amino acids, with the highly conserved WRKYGQK domain at the N-terminus and a zinc-finger motif at the C-terminus. WRKY TFs are DNA-binding proteins that bind the W-box (TTGACT/C), although other binding sites have been reported [65,66].

WRKY TFs represent an active research area in which significant findings have been reported and extensively reviewed. The comprehensive WRKY TF review by Rushton et al [67] covers their involvement in plant processes including development, seed dormancy, germination, and abiotic and biotic stress responses. Two reviews focused on the roles of plant WRKY TFs in response to abiotic stresses, including drought, salinity, heat, cold, and osmotic stress [68,69]. A recent review by Chen et al. [70] provides a current update on WRKY TFs in model plants and important crops, along with a discussion about how high-throughput technologies have accelerated the study of WRKY TFs. Despite the functional characterization of many WRKYs in model plants and major crops, WRKY TFs in sorghum have not garnered much attention. Here, we describe the current status of the related research in sorghum and briefly touch upon the studies in other important crops, such as maize, rice, and wheat.

The numbers of sorghum *WRKY* genes in different databases are as follows: 134 in the Plant Transcription Factor Database (http://planttfdb.cbi.pku.edu.cn/index.php?sp=Sbi), 97 in the Joint Genome Institute (JGI) annotation database (https://phytozome.jgi.doe.gov/pz/portal.html), and 94 in grass TFDB (https://grassius.org/grasstfdb.php). In previous studies in both model and non-model plants WRKY, TFs have been reported to improve stress tolerance. Although WRKY TFs are regarded as promising candidates for enhanced tolerance to abiotic and biotic stress in major crop species, very little is known about *WRKY* genes in sorghum. Thus, the identification and characterization of sorghum *WRKY* genes will provide the necessary foundation for stress tolerance improvement in this crop species. For comparison, the numbers of *WRKY* genes identified in different crops based on genome-wide approaches include 107 *WRKY* genes in wheat [71], 119 in maize [72], 70 in chickpea [73], and 103 in rice [74].

Numerous studies have established that WRKY TFs regulate diverse processes and control gene expression through combinations of positive and negative regulation [75]. In Arabidopsis, *WRKY13* overexpression enhances cadmium tolerance, attributable in part to WRKY13 binding to the *PDR8* gene; the resulting transcriptional regulation leads to a decrease in cadmium accumulation and thus enhanced tolerance [76]. In contrast, *WRKY12* negatively regulates cadmium tolerance. WRKY12 represses *GSH1* expression by directly binding to the W-box in cis-element, and *WRKY12*-overexpressing Arabidopsis plants have reduced cadmium tolerance [77]. Similarly, *ZmWRKY17* overexpression in Arabidopsis reduces salt stress tolerance and decreased sensitivity to ABA by regulating ABA-dependent and stress-responsive genes [78]. Certain WRKY TFs are known to be involved in the response to drought and heat stress, and studies in the last few years have identified *WRKY* genes responsible for drought tolerance in different species. He et al. [79] reported that overexpressing wheat *WRKY1* and *WRKY33* in Arabidopsis enhanced drought and heat tolerance in the transgenic plants. Also, *ZmWRKY40*-overexpressing transgenic Arabidopsis exhibited significant drought tolerance, through the regulation of ROS and stress-related genes [80]. Beyond the functional characterization of *WRKY* genes in the model plant Arabidopsis, the functions of many WRKY genes remain to be validated in non-model species. Analysis of *TaWRKY2* overexpression in wheat by Gao et al. [81] confirmed that it enhanced drought tolerance and also showed that it increased yield in the transgenic wheat. Similarly, transgenic wheat overexpressing Arabidopsis *WRKY30* exhibited drought and heat tolerance, and expression analysis revealed that stress-related genes and enzyme-encoding genes were induced [82]. These findings highlight the important functions of WRKY TFs and raise the prospect of their potential for crop improvement in terms of stress tolerance.

Several recent reports have established that WRKY TFs play significant roles in the regulation of defense responses to pathogen attack. Peng et al. [83] reported that rice *WRKY80* and *WRKY4* enhance resistance to sheath blight disease in rice: WRKY80 binds to the W-box in the promoter region of *WRKY4*, which acts as a positive regulatory circuit for rice resistance against *Rhizoctonia solani*. Using expression analysis, Sureshkumar et al. [84] identified *WRKY7*, *WRKY58*, *WRKY62*, *WRKY64*, and *WRKY76* as highly expressed genes under rice blast disease. In another study, overexpression of rice *WRKY67* enhanced the resistance of transgenic rice plants to two major diseases in rice, blast, and bacterial blight, making *WRKY67* an ideal target for rice improvement [85]. In wheat, Wang et al. [86] reported that *TaWRKY62* silencing increased susceptibility to stripe rust disease, whereas *TaWRKY49* silencing enhanced resistance to the disease relative to non-silenced plants. The enhanced resistance resulting from *TaWRKY49* silencing was linked to genes responsive to jasmonic acid and salicylic acid, *TaPR1.1* and *TaAOS*, and the ROS-related genes *TaCAT* and *TaPOD*, whose expression levels were all suppressed. In contrast, reduced resistance following *TaWRKY62* silencing was related to the high expression of *TaPOD* and *TaPR1.1* and suppression of *TaPIE1* and *TaAOS* [86]. WRKY TFs also involved in the responses to plant viruses, for example, *WRKY8* participates in the response to crucifer-infecting tobacco mosaic virus (TMV-cg) in Arabidopsis by regulating both the ABA and ET signaling pathways to confer resistance against the infection [87] Finally, WRKY TFs may also mediate cross-talk between abiotic and biotic stress response pathways. For example, Lee et al. [88] found that *OsWRKY11* can enhance pathogen defense and drought tolerance in rice by modulating abiotic and biotic stress-related genes.

Although a decade has passed since the sequencing of the sorghum genome was completed, very little is known about WRKY TFs in sorghum, in contrast to other crops, in which many *WRKY* genes have been functionally characterized. Therefore, increased attention and study are needed for sorghum *WRKY* genes. Increasing progress is being made in important crops such as maize and wheat, representing a shift from research in model plants (Arabidopsis) to crop species. The reported findings provide extensive data about the functional characterization of WRKY TFs under abiotic and biotic stresses. Nevertheless, to actualize the promising potential of *WRKY* genes for crop improvement, future analyses will need to be performed under field conditions to detect the actual changes in phenotype, agronomic traits, and yield.

## 6. ERF/DREB TFs

Ethylene-responsive factor/dehydration-responsive element-binding (ERF/DREB) proteins make up a large TF subfamily belonging to the APETALA 2/ethylene-responsive element binding factor (AP2/ERF) TF family, which was first identified in Arabidopsis [89,90]. The ERF/DREB subfamily includes stress-inducible factors, and many of these genes are known to be involved in biotic and abiotic stress responses [34]. ERF/DREB TFs play a vital role in ABA-independent pathways, which regulate stress-induced genes. ERF regulates gene expression by binding to the ethylene-responsive element via the GCC box (AGCCGCC) in the *cis*-element of target genes, and DREB binds to the (DRE/CRT) *cis*-acting dehydration-responsive element/C-repeat (TACCGACAT) [89]. These subfamilies are well studied, and many ERF and DREB members have been characterized and reviewed. For example, Mizoi et al. [91] discussed their major functions, with an emphasis on DREB TFs involved in abiotic stress responses.

The complete genome sequences of many crops are now available, and AP2/ERF members have been identified in many species, including 105 *ERF* genes in sorghum [92], 171 in millet [93], and 118 in barley [94]. The expression analysis in sorghum revealed that *SbDREB2A* and *SbDREB2B* are upregulated under salt and cadmium stress. Furthermore, the miRNAs have been detected in the target regions of *SbDREB2* genes which indicating its roles in post-transcriptional regulation [95]. Transgenic rice plants overexpressing sorghum *DREB2* exhibited tolerance to water deficits, and the yield improved in transgenic rice compared to wild type [96]. Using a gene stacking approach in rice, Kudo et al. [97] found that overexpression of both *DREB1A* and PHYTOCHROME-INTERACTING FACTOR-LIKE1 rice (*OsPIL1)* improved drought tolerance in transgenic plants, similarly to the *DREB1A* overexpressor. Notably, abiotic stress-responsive *DREB1A* downstream genes and *OsPIL1*-related genes were increased in the double overexpressor. In various model plants and crops, DREB/C-repeat factors (CBFs) have also been found to play an important role in increase stress tolerance, particularly in response to cold stress. In sorghum, transcriptome analysis revealed that DREB/CBFs were differentially regulated in response to cold stress in cold stress genotypes [98]. Similarly, transcriptomic analysis of chilling-tolerant Chinese sorghum showed chilling-induced upregulation of cold-response regulator CBF TFs and genes involved in this regulatory [99].

Transgenic wheat plants overexpressing *TaERF3* exhibited significant tolerance to drought and salt stress [100]. In contrast, plants with virus-induced gene silencing (VIGS) of *TaERF3* were more sensitive to salt and drought compared to unsilenced plants, and *TaERF3* was found to regulate other stress-related genes. Recently, El-Esawi and Alayafi. [101] reported that transgenic cotton plants overexpressing *StDREB2* from potato exhibited enhanced drought tolerance. Furthermore, the expression levels of stress-induced genes, such as *GhERF2*, *GhDREB1B*, and *GhDREB1A*, and antioxidant genes were higher in the transgenic cotton plants compared to wild type cotton, indicating that *StDREB2* overexpression might enhance drought tolerance through the upregulation of genes that mediate defense mechanisms. A recent study showed that the expression of two *DREB/CBF* genes (*TaDREB3* and *TaCBF5L*) in transgenic wheat and barley could be modulated by the stress-responsive promoters *HDZI-3* and *HDZI-4*, which are induced by drought and cold. The expression of both genes under these promoters improved tolerance to drought and frost in transgenic barley, and frost tolerance in wheat seedlings [102].

In addition to their roles in abiotic stress tolerance, ERF/DREB factors may also have conserved functions important for the regulation of disease resistance pathways. For example, expression profiling of grape *VvERF* showed that it was highly expressed in response to *Botrytis cinerea* infection [103]. Also, transgenic Arabidopsis overexpressing the *VaERF20* gene from Amur grape had enhanced resistance to *Pseudomonas syringae* and *Botrytis cinerea* pathogens [104].

The findings summarized above indicate that ERF/DREB TFs have important roles in response to both abiotic and biotic stresses. These previous studies in different plant species provide a solid foundation for the application of ERF TFs to improve plant stress responses.

## 7. bZIP TFs

Basic leucine zipper (bZIP) TFs make up one of the largest TF families. The highly conserved bZIP dimerization domain comprises a basic region and a less conserved leucine zipper domain. The basic region contains a nuclear localization signal followed by approximately 16 amino acid residues with an invariant N− -x_7_-R/K motif that binds to DNA; the leucine zipper region is responsible for the dimerization ability of bZIP [105,106]. Previous reports have shown that bZIP proteins bind specifically to C-box (GACGTC), A-box (TACGTA), G-box (CACGTG), PB-like (TGAAAA), and GLM (GTGAGTCAT) sequences in the *cis*-element of stress response genes [106]. The different numbers of bZIP members that have been identified in different plants include 92 in sorghum [14], 78 in Arabidopsis [107], 191 in wheat [105], 125 in maize [72], and 63 in sesame [108]. Recently, genome-wide analyses have also been used to identify bZIP TFs in a wider range of species, including cassava, watermelon, and peanut [109,110,111]. bZIP TFs are associated with different biological activities, including the response to biotic and abiotic stresses [112,113].

In different species, transgenic overexpression of *bZIP* genes has been shown to enhance tolerance to abiotic and biotic stresses. For instance, transgenic Arabidopsis plants overexpressing wheat *TabZIP60* had significantly improved tolerance to salt, drought, and freezing stresses [114]. The study further showed that *TabZIP60* binds to ABA-responsive *cis*-acting elements (ABREs) of ABA-responsive genes. Similarly, overexpression of the novel wheat gene *TaZIP14-B* in Arabidopsis conferred tolerance to salt, freezing, and ABA in the transgenic plants [115]. It has been reported that bZIP TFs play a vital role in the ABA signaling pathway. One notable example is the rapid expression increase of the ABA-induced genes *LEA* and *Rab16* in rice plants overexpressing *OsbZIP42* [116]. *OsbZIP42* overexpression also enhanced the tolerance of the transgenic plants to drought stress, potentially due to the changes in ABA signaling and ABA-dependent modification. bZIP TFs can also play a negative regulatory role by modulating ABA signaling. The expression analysis by Pan et al. [117] showed that the tomato *SlbZIP38* gene was downregulated under drought, ABA, and salt stress. Moreover, transgenic tomato plants overexpressing *SlbZIP38* had reduced drought and salt tolerance, providing further evidence of the negative regulatory role of *SlbZIP38*.

Chang et al. [118] used a multigene assembly approach (co-overexpression of two genes) and reported that the overexpression of a constitutively active form *OsbZIP46CA1* significantly enhanced drought tolerance in transgenic rice. Rice *OsbZIP66* is associated with drought tolerance, and the expression analysis by Yoon et al. [119] showed that expression of the gene was induced by drought, salt, and ABA. Moreover, transgenic rice plants overexpressing *OsbZIP66* had enhanced drought tolerance, which was positively correlated with the expression level of *OsbZIP66*. In another study, overexpression of the *EcbZIP17* gene from finger millet in transgenic tobacco plants conferred enhanced tolerance to abiotic stresses and improved plant growth and yield [120].

A few studies have indicated that bZIP TFs are involved in plant responses to pathogen attack. The review by Alves et al. [112] covered the responsiveness of plant bZIP TFs to pathogens and also summarized the interacting partners of bZIPs at the molecular level and the signal transduction pathway during pathogen infection. The review by Noman et al. [121] described plant bZIP TF responses to different pathogens and summarized the known functions of bZIP TFs as positive or negative regulators of resistance to pathogen infection. A research article by Alves et al. [122] described four *bZIP* genes in soybean (*GmbZIPE1*, *GmbZIPE2*, *GmbZIP105*, and *GmbZIP62*), and the findings indicated that bZIP proteins contribute to the defense response against Asian soybean rust disease (ASR) through the expression regulation of ASR-related genes. Expression analysis in cassava revealed that *MebZIP3* and *MebZIP5* were induced by salicylic acid, hydrogen peroxide, and *Xanthomonas axonopodis* Pv. *manihotis* [123]. Additionally, transgenic tobacco overexpressing *MebZIP3* and *MebZIP5* had improved resistance against cassava bacterial blight disease. In contrast, silencing *MebZIP3* and *MebZIP5* in transgenic plants reduced the transcript levels of the defense-response genes and led to a disease-sensitive phenotype. Some bZIP TFs play multiple roles and are associated with both biotic and abiotic stresses. For example, transgenic Arabidopsis plants overexpressing *CabZIP* from pepper showed enhanced tolerance to *Pseudomonas syringae* Pv. *tomato* DC3000 as well as drought and salt tolerance [124]. Despite the available research showing the involvement of bZIP TF genes in different abiotic and biotic stress responses, many of these genes have yet to be characterized in sorghum. These findings would ultimately help improve the tolerance of sorghum to stress conditions. Additionally, the available research findings on bZIP TFs involved in plant responses to insects and other biotic stresses are relatively limited. Therefore, future studies will need to elucidate the functions of bZIP TFs in response to nematodes and other pests because of their destructive impact on many important crops.

## 8. Others

In addition to the examples of TF functional characterization described above, many other TFs from different families contribute to abiotic and biotic stress responses. For example, heat shock (Hsf) TFs are induced in response to a large number of different stresses. Transgenic Arabidopsis overexpressing *ZmHsf05* exhibited enhanced tolerance to heat stress, indicating that *ZmHsf* plays a vital role in heat response [125]. In sorghum, Gomes et al. [126] identified 25 *SbHsf* genes using a genome-wide approach, some of which were differentially expressed under drought, cold, and salt stress. Another important group of TFs is the bHLH family, and a recent analysis identified 208 bHLH members in maize [127]. Overexpression of Arabidopsis *bHLH122* led to resistance to drought, osmotic stress, and salt stress, and microarray analysis showed that *bHLH122* affected the expression of numerous abiotic stress-related genes [128]. Thus, *bHLH122* was found to have a positive role in the response to drought, salt, and osmotic stress. Homeodomain-leucine zipper (HD-Zip) TFs similarly have diverse functions in mediating stress response. Moreover, mutations in many genes encoding HD-Zip proteins from several species have been associated with improved agronomical phenotypes [129]. TFs influences the expression of thousands of genes, and the responses of TFs to ABA provide a versatile model for the examination of transcriptional regulation using ChIP-Seq. Song et al. [130] identified 21 ABA-related TFs in Arabidopsis; also, the determinants of combined dynamic TF binding along with hierarchy among TFs were defined. Moreover, new family of transcription regulators that exhibited altered expression in response to ABA and salt stress, and the authors proposed that the ABA response is orchestrated by a handful of master regulators, including TFs. Stress-associated protein (SAP) is also recognized as a key player in multiple stress responses. For example, transgenic plants overexpressing *AtSAP13* had significant tolerance to drought, salt, and heavy metals (Zn, Cd, and AsIII), and dozens of AP2/EREBP TFs were found to interact with the *AtSAP13* promoter [131]. However, many other TF families have yet to be functionally characterized in the context of stress response, such as the homeobox (HB), DNA binding with one finger (Dof), LATERAL ORGAN BOUNDARIES (LBD), and plant-specific WUS homeobox (WOX) families. Overall, the findings in the literature show that many TFs related to stress response have been identified in different species. Nevertheless, there are many TFs that have yet to be identified along with known TFs whose functions are not yet known.

## 9. Non-Coding RNAs Regulate Gene Expression

Non-coding RNAs (ncRNAs) have wide-ranging impacts and have become a major research area. Plant genomes encode millions of ncRNAs, some of which play vital roles in regulating gene expression at the transcriptional and post-transcriptional levels [132]. NcRNAs are classified into different groups based on their length and function. Small RNAs are 20–30 nucleotides in length and include small interfering RNAs and microRNAs (siRNAs and miRNAs). Two other groups are medium-length RNAs, which are 50–200 nucleotides in length, and long ncRNAs (lncRNAs), which are >200 nucleotides in length [133]. LncRNAs have garnered much attention, and they are known to have crucial roles in response to abiotic and biotic stresses [134]. Advanced technologies that facilitate genome-wide study, such as RNA-sequencing and next-generation sequencing, have been critical for the identification of lncRNAs in many species, including Arabidopsis, wheat, rice, and maize [135,136,137,138]. Boerner et al. [132] reviewed lncRNA identification and characterization as well as the mechanisms of microRNAs (miRNAs) and lncRNAs in response to different stresses in plants. Despite their comprehensive summary and explanations of both lncRNA and miRNA functions in response to plant stress, they underscored that more studies are needed to more fully understand the regulatory roles and signaling pathways of ncRNAs in response to stresses. The following paragraphs summarize recent updates on small RNAs and lncRNAs in response to abiotic stresses and current studies investigating these plant responses to disease stress.

Several regulatory small RNAs play important roles in biotic stress response by controlling the expression of TFs. Specifically, these small RNAs modify transcription regulators such as WRKY, NAC and zinc-finger proteins in Arabidopsis through post-transcriptional silencing and signaling via reactive oxygen species [139]. miRNAs and TFs are master regulators, and the combinatorial regulation mediated by TFs and miRNAs guides the proper progression of biological events [140]. In rice, NAC TF genes targeted by miR164 negatively regulate drought tolerance [141]. A MYB TF that is upregulated under heat stress was found to be targeted by miR828 and miR858. In wheat, the regulation of NAC21/22 by miR164 affects the resistance to stripe rust disease [142]. The types of small RNAs (sRNAs) that regulate gene expression in plants include miRNAs and natural antisense miRNAs (nat-miRNAs), and miRNAs have been shown to be involved in the responses to different abiotic and biotic stresses. For instance, in sorghum, 526 novel miRNAs were identified, and 96 were found to be drought-responsive miRNAs [143]. In tomato, stress-responsive miRNAs were recently identified, and 10 miRNAs were found to have stress-specific expression responses [144]. miRNAs are ideal candidates to investigate the crosstalk between gene expression networks and signaling pathways, because many miRNAs target TFs that are master regulators of gene expression [145].

In Arabidopsis, a novel drought-induced lncRNA (DRIR) was identified as a positive regulator of plant responses to salt and drought stresses [146]. Expression profiling revealed that *DRIR* expression could be significantly induced by drought, salt, and ABA stresses, and its levels were lower under normal conditions. In addition, transgenic plants overexpressing Arabidopsis *DRIR* had increased drought and salt tolerance.

Using genome-wide transcription analysis in maize, Li et al. [147] identified 1535 drought response lncRNAs. Furthermore, structural characterization showed that lncRNAs are less complex and shorter than protein-coding genes and also have fewer exons. Also, lncRNAs exhibit different expression patterns at different growth stages and in response to drought stress. For example, at the reproductive stage, lncRNAs have an altered expression pattern, and they are more abundant under drought stress. As such, the study by Li et al. [147] provides new insight into lncRNAs involved in gene regulation in the drought response. Similarly, a research study by Nagaraju et al. [148] used deep RNA-sequencing and identified 1710 lncRNAs responsive to salt and boron stress as well as 848 stress-responsive potential trans-natural antisense transcript (NAT) lncRNAs, which were associated with transcription regulation, response to abiotic stimulus and response to stress. These ncRNAs could potentially regulate genes related to transcriptional regulation, abiotic stress response, and other biological functions. In Arabidopsis, NAT-lnRNAs were found to acts in *cis* to activate *MAF4* expression during cold stress. The different expression profiles of *MAF4* revealed the vital role of lncRNAs in coordinating the vernalization response and in other biological processes [149]. In cassava, systematic screening identified 318 lncRNAs responsive to cold and drought, and they were generally coexpressed concordantly or discordantly with their neighboring genes [150].

Several lncRNAs have been implicated in the regulation of genes in response to biotic stress such as pathogens. Thousands of lncRNAs were identified in rice lines resistant to the rice blast pathogen *Magnaporthe oryzae* [151]. Functional classification of these lncRNAs showed that the numbers of long intergenic ncRNAs were higher in the resistant rice lines compared to the numbers of antisense lncRNAs. Overall, the findings suggested that lncRNAs have a role in regulating resistance genes in rice during *M. oryzae* infection [151]. Perotti et al. [152] identified lncRNAs involved in the resistance to fungal disease (*Verticillium dahliae*) in cotton. The conclusions about the lncRNA functions were based in part on comparisons of the lncRNA expression patterns in disease-resistant cotton (*Gossypium barbadense*) and susceptible cotton (*G. hirsutum*), which showed clear disease response mechanisms. Interestingly, some lncRNAs may function in multiple stress responses. For example, 125 lncRNAs were found to respond to powdery mildew disease infection and heat stress [135]. The findings from the research study by Dixit et al. [153] included 565 lncRNAs involved in the response to root-knot nematode; 15 lncRNAs were studied further and were found to have different expression patterns.

In summary, remarkable progress has been made in the last few years although studies on small RNAs and lncRNAs in sorghum are limited. Besides TFs, epigenetic manipulation of gene expression is another potential strategy for improving plant responses to abiotic and biotic stresses.

## 10. Concluding Remarks and Future Perspectives

Considering the world’s population is expected to reach 9 billion by 2050 [154], new strategies, such as improved stress tolerance in crops, are critical for increasing productivity to satisfy the projected food and energy demands of the world population. As key stress-tolerance mediators, TFs could be manipulated to boost the tolerance of different crops to abiotic and biotic stresses. In the last two decades, there has been significant progress in the identification and characterization of major TF families, such as WRKY, NAC, MYB, DREB, and bZIP, in response to both abiotic and biotic stresses. In this review, we summarized recent reports about stress-induced TFs in the top five cereal crops (wheat, rice, maize, barley, and sorghum), and their potential for sorghum improvement. However, the discussed TF families have been extensively studied in the major crops, in sorghum, the studies are limited on some TF families such WRKY TFs.

Recent reports have shown that TFs are now being validated in both model and non-model plants. However, the majority of these studies are performed in the laboratory, and there is a dire need to test candidate TF genes that may improve tolerance in transgenic plants under field conditions. As shown in the literature, TF responses to stress conditions can be extremely complicated. The overexpression of a single TF gene might promote or suppress a vast array of downstream genes, a single gene might be regulated by different TFs that bind to its *cis*- elements, and a single TF might respond to different stresses, For example, *OsNAC6* was found to improve tolerance to high salt, dehydration, and blast disease in transgenic rice plants. Moreover, *OsNAC6* activates the expression of genes that are induce by biotic and abiotic stress, such as *AK104277* (peroxidase gene) and *AK110725* [25]. In Arabidopsis, 34 TF families are known to contain 1533 TFs; interestingly, 45% of these TFs are specific to plants [22]. It would be a tremendous challenge to individually characterize millions of TF genes at the molecular level. Therefore, future studies may undertake combinatorial approaches to study multiple TFs and multiple stresses to elucidate the cross-talk among different TFs, as opposed to a single TF and stress. The availability of complete genome sequences for an increasing number of species and breakthroughs in sequencing technology have facilitated the identification and characterization of TFs. Moreover, published genome databases should enable in silico analysis of genome-wide annotation data for the identification of TFs. For example, the *ERF* gene family was identified using sorghum genomic databases [92]. Genomic identification strategies have benefited tremendously from technologies such as chromatin immunoprecipitation with massively parallel sequencing (CHIP-Seq) and next-generation sequencing (NGS). CRISPR/Cas9 (clustered regularly interspaced short palindromic repeats/CRISPR-associated protein 9) gene editing is a powerful tool that could be used to modulate TFs with the goal of improving plant stress tolerance. A recent publication by Bao et al. [155] provides a comprehensive review on the application of CRISPR/Cas9 for evaluating genes and gene functions involved in environmental stress tolerance or resistance, crop improvement, and the transcriptional and translational regulation of the genes. Future studies will need to address TF gene functional redundancy. Additionally, although previous analyses of TF gene overexpression in response to a specific stress have been highly informative, studies focusing on crop yield and productivity are needed; that is, future investigations will need to address whether the overexpression of stress-related TF genes in transgenic plants improves their stress tolerance and growth and also determine whether there is a negative effect on yield under field conditions.

Numerous reports have confirmed and characterized the involvement of TFs in response to abiotic and biotic stresses, but the molecular mechanisms of many of these TFs are not understood. Plant TF responses to abiotic and biotic stresses are extremely complex, and several TF families are clearly associated with single or multiple stresses along with complex cross-talk between different signal transduction pathways. Collectively, the findings from previous reports indicate the potential application of TF genes to enhance stress tolerance/resistance in important crops; however, extensive studies are needed to understand the mechanisms of these TFs.

## Figures and Tables

**Figure 1 genes-10-00771-f001:**
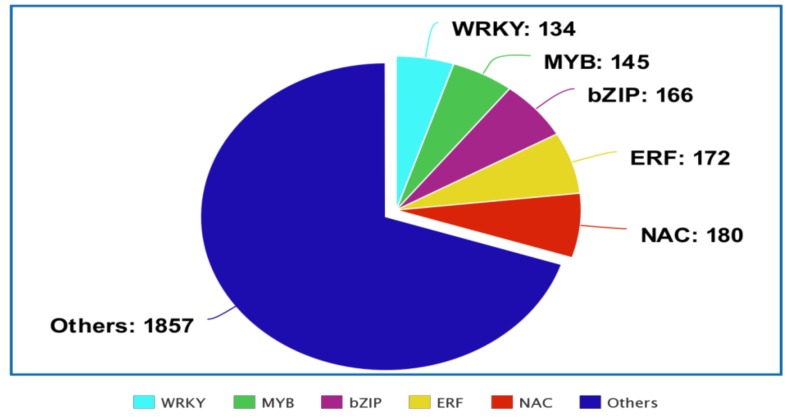
Pie chart showing the number of transcription factor (TF) genes in NAC, MYB, ERF, bZIP, WRKY, and other families in sorghum.

**Figure 2 genes-10-00771-f002:**
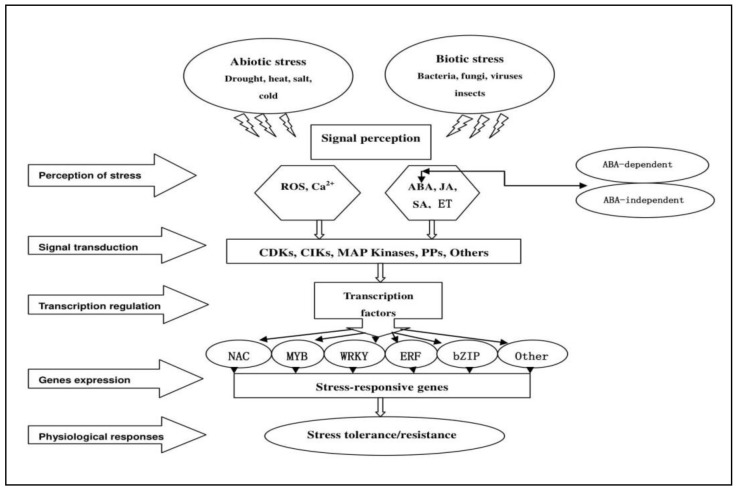
A schematic model of the signaling pathways involved in abiotic and biotic stress responses.

**Table 1 genes-10-00771-t001:** Plant transcription factor databases.

Transcription factor databases	Website
Plant Transcription Factor Database	http://planttfdb.gao-lab.org/
MOROKOSHI Sorghum Transcriptome Database	http://sorghum.riken.jp/morokoshi/Home.html
Grass Transcription Factor Database	https://grassius.org/grasstfdb.php
iTAK-transcription Factor Database	http://itak.feilab.net/cgi-bin/itak/index.cgi
Sorghum Functional Genomic Database	http://structuralbiology.cau.edu.cn/sorghum/index.html
Phytozome database	https://phytozome.jgi.doe.gov/pz/portal.html#
PlnTFDB	http://plntfdb.bio.uni-potsdam.de/v3.0/

**Table 2 genes-10-00771-t002:** General features of the discussed transcription factor (TF) families.

TF Family	DNA-binding Domain	*Cis*-acting Element	Structural Features
**NAC**	NAC domain	NACRS (TCNACACGCATGT)	NAC domain consist of 150 amino acids residues in N-terminal and variable transcription regulatory in their C-terminal.
**MYB**	MYB domain	MYBR (TAACNA/G)	MYB domain composed of multiple repeats each repeat about 52 amino acids which forming a helix–turn–helix (HTH) structure.
**WRKY**	WRKYGQK domain	W-box (TTGACT/C)	WRKY domain is ~60 amino acid residues in length, and also has a zinc-finger structure at the C-terminus which either Cx_4-5_Cx_22-23_HxH or Cx_7_Cx_23_HxC.
**ERF/DREB**	AP2/ERF domain	GCC box (AGCCGCC) and (TACCGACAT)	Composed of 60 amino acids with conserved domain consist of Three parallel β-sheets and putative amphiphilic α-helix.
**bZIP**	bZIP domain	C-box (GACGTC), A-box (TACGTA), G-box (CACGTG), PB-like (TGAAAA), and GLM (GTGAGTCAT)	bZIP domain consist of ~16 amino acid residues which containing nuclear localization signal followed by an invariant N-x7-R/K motif contacts the DNA.

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
