# Peer review of "Transcription Factors Associated with Abiotic and Biotic Stress Tolerance and Their Potential for Crops Improvement"

_genes, 2019, doi:10.3390/genes10100771_

Round 1

Reviewer 1 Report

This review on stress-associated Transcription Factors in crops is very well written and organized. I am not familiar with all the literature cited, but the manuscript seems to be a comprehensive and detailed summary of the current literature in this field. Overall, the article is a descriptive summary, with only a very short discussion of potential disadvantages and trade-offs in using TFs for crop improvement.

In order to help readers to keep an overview, a list of abbreviations would be very helpful, also explaining gene names. Similarly, a table summarizing the main features of the discussed transcription factor families would be helpful. 

Reviewer 2 Report

Baillo et al summarized the role of plant transcription factors (TF) involved in Abiotic and biotic stress tolerance highlighting Cereal crop Sorghum. The review gave a brief introduction of TF families provided by TF databases and the general regulatory pathway of plant TFs response to stress, focused on the progress in  NAC,  MYB, WRKY, ERF/DREB and bZIP TFs, Finally, the authors provided some future perspectives. In general, the outline of this review are reasonable. Bellow are some suggestions for  improving the manuscript.

Biotic stress also includes virus,  therefore, virus should be included in line 37 and Figure 2. And in the WRKY section, some reference with WRKY involved in virus response should be mentioned.  TFs can act as both activators and repressers, therefore, in Figure 2, “Stress-responsive genes” instead of “Stress-induced genes” should be showed as the downstream of TFs.  In the section of “Others”, the 21 TFs responsive to ABA published in 2017 Science (A transcription factor hierarchy defines an environmental stress response network) should be discussed.  The section of “Non-coding RNAs regulate gene expression” show no connection with the topic of the review focusing on TFs.  Small non-coding RNAs (like miRNAs) that regulate TFs involved in abiotic and biotic stress were encouraged to be listed. Also, the stress-responsive lncRNAs that regulate TFs should be highlighted. For instance, a cold-induced NAT-lncRNA that regulate MAF 4 (MAD-box TF family) published in Nature communications (2018) is a good example.

Reviewer 3 Report

Major point

The authors have summarized results of a number of researches in which various types of TFs are overexpressed (or, sometimes, silenced) in homologous or heterologous systems. This paper may provide a brief overview on the present status of this research field; the content is, however, almost mere enumeration of research results, mostly concerned with plants other than sorghum, and thus far from what is expected from the title. The authors may also reconsider whether ncRNAs should be included in the category of TF. It may be better to reconsider the logical structure of the article so that the collected research results should suggest, more concretely, something that is valuable for planning lines of future research and/or improvement in sorghum.

Minor points

Introduction

Figure 1; The alignment sequence of the TF types may be changed to clock-wise.

  The regulatory functions of plant TFs in response to abiotic and biotic stresses

Figure 2; Information shown in the figure should be explained more clearly; the correspondence to the description in the main text should also be improved.

  NAC TFs

Line 133, Why “interestingly”?

Line 146-147, What are “several physiological changes” and “downstream genes”?

Line 149, implicating involvement of TaNAC29?

Line 176, What are “cellular component factors”?

Line 180, The results…of what?

Line 186, Are there some structural or phylogenetic difference among TFs acting positively and those acting negatively?

Line 189, stress response genes…indirectly via ABA? directly?

MYB TFs

Line 219, 1R-MYB, R2R3-MYB, 3R-MYB, 219 and 4R-MYB; More explanation is necessary.

Line 230, active factors in abiotic stress signaling because…; Meaning unclear.

Line 248, What are “early signaling, late-response genes”

Line 270, “lignin synthesis accumulation” sounds strange. Synthesis? Accumulation? Synthesis and accumulation?

Line 273, shed light on the mechanisms; how?

  WRKY TFs

Line 299, Although WRKY TFs are regarded as promising candidates for enhanced tolerance to abiotic and biotic stress in major crop species; Why?

Line 303, For comparison; Why numbers only are compared at this point?

Line 312, ZmWRKY17 312 overexpression in Arabidopsis decreases ABA by regulating ABA-dependent and stress-responsive genes; why and how does ABA decrease?

Line 322; enhanced drought stress?

ERF/DREB TFs

Line 361; These subfamilies?

Line 367, “SbDREB2 is involved in post-transcriptional regulation”; How can a TF be involved in post-transcriptional regulation?

Line 370; explain OsPIL1.

Line 378, Furthermore,…; So, what is implicated?

Line 383, “overexpression of the cucumber ERF004 gene increased its expression level”; unnecessary.

bZIP TFs

Line 414; What determines positive or negative regulatory role of a given bZIP TF?

Line 419; what is multi-gene assembly approach?

Non-coding RNAs regulate gene expression

Line 460, mechanisms in response to stresses in plants; mechanism of what?

Line 473, Nevertheless; why “nevertheless”?

Line 479, trans-natural antisense transcript (NAT) lncRNAs; define.

Line 482, “they were generally co-expressed consistently or inconsistently with their neighboring

genes [130]”; what does it mean?

Line 497-, “Epigenetic manipulation of gene expression is another potential strategy for improving plant responses to abiotic and biotic stresses”; Are ncRNAs “transcription factors”?

Others

Line 504, and the expression of Hsf was upregulated in these transgenic plants; unnecessary explanation for overexpression line.

Concluding remarks and future perspectives

Line 528-, “many other TF families have yet to be characterized in the context of stress response, including the homeobox (HB), DNA binding with one finger (Dof), LATERAL ORGAN BOUNDARIES (LBD), and plant-specific WUS homeobox (WOX) families.” These TFs are not mentioned in the main text, even though there ore “ 9. Others” section. It seems strange to me.

Line 532, “with a particular emphasis on findings in sorghum”; the description concerning sorghum is so limited that the title appears inappropriate.

Line 537-, “For example, the overexpression of a single TF gene might promote or suppress a vast array of downstream genes, a single gene might be regulated by different TFs that bind to cis-acting elements in its promoter, and a single TF might respond to different stresses.”; no clear example is not presented.

Line 542-, “future studies may undertake combinatorial approaches to study multiple TFs and multiple stresses to elucidate the cross-talk among different TFs, as opposed to a single TF and stress.”; If so, the authors should try to demonstrate the possibility of such an approach in this review.

Line 558-, “whether the overexpression of stress-related TF genes in transgenic plants improves their stress tolerance and growth and also determine whether there is a negative effect on yield.”; What does previous studies suggest about this issue? Nothing?

Reviewer 4 Report

In this review Baillo et al present a broad description of the major TF families in several plant species in relation to their roles in stress response. While the stated emphasis of this review is sorghum, the conclusion of most sections seems to be that little is known in sorghum? There was more discussion of other species - which is fine - although this discussion could have been better integrated with the emphasis on sorghum. For instance, are all the TFs discussed in other species also found in sorghum? If they are not found in sorghum are they good targets to be transgenically expressed in sorghum - the relevance could be discussed more clearly. That said, the descriptions in this review provide interested researchers with an overview of the different TFs with many examples of functional characterisation and efforts to apply this knowledge to crop improvement. 

Below are some points that I think would improve the manuscript:

- Line 57 “Transcription factors (TFs) modulate gene expression by binding to cis-acting elements in gene promoters under different biological contexts”. Here (and throughout the manuscript) the authors should mention that TFs also bind to distal cis-elements not just promoters, especially in crops with large genomes: these can be 100kb away from the promoter! Some recent papers on this topic should be cited.

- Line 61 What defines a TF? The authors might consider adding a definition to give context to this discussion.

- Line 449 it would be more relevant to discuss some miRNA examples, which have clear functions in gene regulation and stress response, rather than dedicating this section mostly to lncRNAs. 

- Line 519 “there are many TFs that have yet to be identified”. What does this mean? Is this suggesting that in sequenced and annotated genomes there are proteins of unknown function that are actually TFs???

- Line 562 “the regulatory mechanisms of many of these TFs are not understood”. Can the authors elaborate on this statement, it is not clear what is meant? Does this refer to the molecular function of the TF or the broader regulatory pathway that they are part of?

Reviewer 5 Report

Baillo et al. review on transcription factors (TF) associated with stress tolerance, with focus on sorghum, is interesting. They covered major TFs (NAC, MYB, WRKY, DREB, and bZIP) and their role in protecting sorghum, maize, rice, wheat, and barley. Overall, the review is very informative and can help/guide future work on transcription factors.

There are some comments to improve the manuscript.

CBF/DREBs are canonical cold tolerance genes in model plants. Include their role in conferring cold tolerance in tropical crops. Here are two transcriptome studies showing CBFs were differentially regulated in chilling-tolerant Chinese sorghums [Chopra et al. BMC Genomics (2015) DOI 10.1186/s12864-015-2268-8, and Marla et al. (2017) The Plant Genome DOI: 10.3835/plantgenome2017.03.0025]. These two research articles suggest the role of CBFs in cold tolerance in sorghum. Additionally, association studies in rice and wheat also show CBFs as candidate genes for cold tolerance in these crops. References 4 and 5 are not correctly cited. The cost of yield loss is more than 30–50 million dollars per year. Find a good reference and include the appropriate estimated losses. Table 1: Columns1 and 2 are not aligning properly

Round 2

Reviewer 3 Report

The authors have revised the manuscript according to the reviewer's comments. However, the reviewer feel that disagreement between the title and contents of the article, which was pointed out as Major points 1 and 2 in the previous reviewer's comment, still remains.
